# Effect of Crown Layers on Reproductive Effort and Success in Andromonoecious *Aesculus indica* (Wall. ex Camb.) Hook (Sapindaceae) in a Temperate Forest of Garhwal Himalaya

**DOI:** 10.3390/plants13020183

**Published:** 2024-01-10

**Authors:** Priya Pradhan, Arun Sukumaran, Vinod Prasad Khanduri, Bhupendra Singh, Deepa Rawat, Manoj Kumar Riyal, Munesh Kumar, Marina M. S. Cabral Pinto

**Affiliations:** 1College of Forestry, VCSG Uttarakhand University of Horticulture and Forestry, Tehri Garhwal, Pauri 249199, India; priyapradhan1201@gmail.com (P.P.); khandurivp@yahoo.com (V.P.K.); rawatdeepa291@gmail.com (D.R.); manojrayal1509@gmail.com (M.K.R.); 2ICFRE—Bamboo and Rattan Centre, Bethlehem Vengthlang, Aizawl 796007, India; arunsukumaranpn@gmail.com; 3Department of Forestry and Natural Resources, HNB Garhwal University, Srinagar Garhwal 246174, India; muneshmzu@yahoo.com; 4GeoBioTec Research Centre, Department of Geosciences, University of Aveiro, 3810-193 Aveiro, Portugal; marinacp@ua.pt

**Keywords:** andromonoecy, ambophily, breeding system, crown layers, protogyny, reproductive success, staminate flowers, stigma receptivity

## Abstract

The andromonoecy is an unusual sex expression in trees in which an individual plant bears both functionally staminate and hermaphrodite flowers on the inflorescences. This study aims to investigate the effect of crown layers on the floral biology and reproductive effort of *Aesculus indica* (Wall. ex Camb.) Hook. The results revealed that the peak period of anthesis was between 06:00 and 08:00 h of the day. Male flower production was predominantly higher as compared to the perfect flowers on the inflorescences. There was no significant variation between total pollen production in staminate and perfect flowers. Features like protogyny and inter-level asynchrony promote xenogamy; however, intra-level asynchrony results in geitonogamy. Controlled pollination treatments revealed the existence of self-incompatibility in flowers. Pollination syndromes in flowers support ambophily. A trend of consistent improvement in reproductive success from lower canopy layers to upper crown layers in the analyzed trees was recorded. The crown layers have a significant impact on flower production, fruit, and seed set. An increase in male flower production due to the increment in the crown is a mechanism of reproductive assurance as a pollen donor and pollinator recipient and also due to the differential cost of expenditure of reproduction in crown layers. Andromonoecy in *A. indica* promotes self-incompatibility, and there was a tapering trend of reproductive success in the crown layers.

## 1. Introduction

Throughout the evolutionary history of plant species, many species have responded to the changing climate. The species in an ecosystem are so firmly adapted to the long-established climatic pattern that they show alterations even in response to the mildest change in climatic factors [1]. The genus *Aesculus* (family Sapindaceae) includes 13 species, several varieties, cultivars, and natural hybrids distributed across the Northern Hemisphere, mainly in temperate climates. The species are organized into five sections: *Aesculus*, *Calothyrsus*, *Macrothyrsus*, *Parryana* and *Pavia* [2]. *Aesculus indica* (Wall. ex Camb. Hook.) (Indian horse chestnut) is a large tree with a round-shaped canopy that falls under the section Calothyrsus. *A. indica* occurs along the Himalayan lowlands between the elevations of 1000 and 3000 m a.s.l. in the forests and shady valleys across Northern Afghanistan, Pakistan, Kashmir, North India, and Nepal [3]. Another species of this genus, viz. *Aesculus hippocastanum* L., has been reported to have allergic significance to children [4], and in the Middle East, 57.4% of the pollen of *Aesculus* is reported as allergic pollen in the atmosphere [5].

The siliceous, deep, draining, and fertile soils are generally appropriate for *Aesculus* [6]. Karlinski et al. [7] reported that the calcium carbonate-rich, moist, and nutrient-rich soils resulted in flat root systems in *Aesculus,* and shallow but more extensive root systems were observed in poor soils. It is a tree species that has been reported to be capable of surviving under varying soil conditions ranging from nutrient-poor, sandy, or clayey soils, though a soil pH between 6.1 and 7.5 is generally recommended for optimum growth [8,9]. The species *A. indica* is grown mostly as a shade tree for parks, arboreta, college campuses, and home landscapes [10] and is also recommended for buffer strips, indicating its role as an important species for land management in urban areas. Apart from this, the bark of *A. indica* has been reported to have astringent properties and is used as a tonic [11]. The wood is used for making utensils, packing cases, tea boxes, and decoration articles [12].

Both in situ and ex situ conservation programs for various plant species are being implemented the world over, but the success is not as expected owing to several intrinsic constraints related to reproduction [13]. Obtaining basic information about the reproductive biology of a species to understand the mechanism through which it has become endangered or evolved through a changing climate cannot be overlooked [14]. The knowledge of breeding behavior and the ecology of pollination that shape the genetic structure of plant populations is extremely crucial for framing out efficient conservation approaches [15]. Complete flowers have fully developed androecium and gynoecium. Pistillate inflorescences were found infrequently in *A. hippocastanum* and in *A. pavia* L., where two plants out of many hundreds had only functionally female flowers [16]. The flowering biology of this genus is characterized by andromonoecious (having perfect and staminate flowers on the same plant) flowers with a large proportion of functionally staminate flowers. Functionally staminate flowers have a vestigial gynoecium with an undeveloped ovary and stunted style [2]. The proportions of female-fertile flowers in *Aesculus* are 0.1% [17]. The male-female flower ratio of different *Aesculus* species was reported as 1.1% for *A. pavia*, ref. [16]; 1.3 to 9.3% for *A. turbinata* Blume [18]; <5% for *A. californica* (Spach) Nutt. [19]; and 7.0% for *A. sylvatica* L. [20]. Studies on the reproductive biology of *A. indica* are required to understand the variability and relationships among floral morphology, mating systems, and floral visitors under different ecological contexts.

Various kinds of stress imparted by the changing climate have decreased growth, elevated the threat of mortality in a lot of forest tree species, and led to an increase in crown defoliation, but the effect of these environmental stresses on the reproduction behavior of trees remains understudied [21]. Along the same lines, studies regarding the reproductive effort of tree species in various layers of the crown in the present scenario of changing climate in the Himalayan region have not been performed thus far for *A. indica*. Reproductive success may vary among the crown layers of the trees. The possible assumptions include differential allocation of photosynthates in crown layers or the influence of sunlight on fruit and seed sets. The variation in effective pollination and fertilization success in different crown layers is also expected. In *A. turbinata*, panicles from the upper crown (presumably receiving optimum photosynthetically active radiation) were longer and had a greater number of flowers per panicle with a higher sex ratio than those from lower portions of the crown. Furthermore, fruit development depends strongly on photosynthesis and light availability [22,23,24]. Leaves in the upper part of trees shade those present below, making a vertical stratification of light within the tree. Consequently, upper leaves are expected to produce more photosynthates than lower, shaded leaves.

We investigated the floral morphology and biology, pollination, and breeding system of *A. indica* to address the following questions: (1) Do the reproductive success and reproductive potential in terms of fruit and seed set percentages in different crown layers vary? (2) Is there any significant difference between the production of staminate and sexual flowers in the sample trees? (3) What is the functional role of the staminate flower? (4) Is pollen production in staminate flowers and perfect flowers equal or not? (5) What is the main pollination system of the study species? (6) What are the major floral visitors, and how do they take part in reproduction?

## 2. Results

### 2.1. Floral Morphology and Anthesis

The sex expression in *A. indica* is andromonoecious. There was a marked variation in the number of staminate flowers and perfect flowers within the inflorescence (t = 10.60, *p* < 0.05, d.f. = 18). The majority of the flowers were staminate, with the pistil absent or rudimentary. The inflorescence was terminal thyrse, broad at the base, forming a pyramidal shape. The flowers were pedicellate and zygomorphic; the number varied from 341 to 441 per inflorescence, and the number of perfect flowers per inflorescence was 13 to 20. The androecium includes seven stamens (occasionally 6, 8, and 9) that are filiform, tomentose, narrow, oblong, curved upward, and longer than the petals. The gynoecium is hypogynous, simple, and slender, with a sessile ovary having six ovules arranged in three cells with two ovules each. Reproductive phenological observations showed that the flower bud emergence takes place in March, and the fruits appear in the last week of April and fully ripen from October to November. Inflorescences were commenced in March and took 27 ± 2 d for enlargement of buds; individual buds took 8–12 d for complete development (Figure 1); the mean time taken for the complete blooming of one inflorescence was 56 ± 2 d; and the mean number of days for fruit development since bud initiation was 142 ± 5 d. Flowers were opened in an acropetal pattern to be noted from the second week of April to June. The peak period of anthesis was recorded between the morning hours of 06:00 and 08:00 h of the day (Table 1). The effect of the tree-to-tree difference on anthesis was not significant (Table 2, *F* = 3.26, *p* < 0.05); however, there was a significant effect of time on anthesis (Table 2, *F* = 968.55, *p* < 0.05). Anther dehiscence begins after 5–6 h of the anthesis.

### 2.2. Pollen Production

The total flower production in the sample trees has oscillated between 18,804 ± 3611 and 151,906 ± 24,179. Pollen grains per stamen varied from 7597 ± 91 to 7739 ± 169 for staminate flowers and 7590 ± 56 to 7781 ± 170 for perfect flowers. Pollen production between staminate and perfect flowers did not differ significantly (t = 0.261, *p* = 0.05). The mean pollen production per flower, inflorescence, and tree ranged between 53,183 ± 643,222 to 54,177 ± 1189; 18,363,790 ± 476,673 to 23,914,032 ± 524,987; and 1,010,008,499 ± 26,217,045 to 8,078,958,841 ± 97,679,046; respectively.

### 2.3. Stigma Receptivity and Pollen Ovule Ratio

The inflorescence of *A. indica* was protogynous; female receptivity lasts about 5–6 d and shows distinct morphological changes. The stigma looked fresh and pink in color throughout its receptive phase. Eventually, the stigma becomes dry, shrinks, and becomes brown in color, which makes clear the cease of receptivity.

There are six ovules per flower; hence, the P/O ratio varied from 8855 ± 65 to 9078 ± 198 in flowers with developed style and from 8863 ± 107 to 9029 ± 198 in flowers with underdeveloped style.

### 2.4. Pollen Viability and Germination

The percentage of viable pollen per slide ranged from 87.54 ± 4% to 90 ± 0.8% for perfect flowers and 85.3 ± 3% to 89.6 ± 4% for staminate flowers. The effect of time and concentration of solution on pollen germination was found to be significant for perfect flowers (*F* = 48.73, *F* = 18.10) and staminate flowers (*p* < 0.05, *F* = 67.84, *F* = 31.13, *p* < 0.05) (Table 2). The maximum germination of 88.17% for perfect flowers and 88.21% for staminate flowers was observed in a 20% sucrose concentration, followed by 83.33% for perfect flowers and 84.19% for male flowers in a 10% sucrose concentration.

### 2.5. Breeding Systems and Floral Visitors

There was a significant effect of treatments on the fruit set (Table 2, *F* = 13.833, *p* = 0.0156). A small difference in fruit set was observed between the open-pollination and the open cross-pollination. The flowers failed to set fruit under the treatment of obligate self-pollination and apomix, indicating the need for pollen transmission vectors. A diverse array of insects belonging to the orders Hymenoptera, Thysanoptera, and Lepidoptera were observed visiting the flowers of *A. indica* during the 8-day observation period (Table 3). Most visits were observed in the morning hours between 08:00 and 10:00 h. The most frequent visitors to flowers were *Bombus* spp. and thrips. There was little difference in the visitation rates of the insects to staminate and perfect flowers. The inflorescence received most *Bombus* spp. in the morning (06:00–08:00 h) and rarely in the afternoon. Bumblebees tended to visit the outward-facing flowers and the flowers at the base of the panicle first. *Tagiades menaka* Moore was found more frequently in the evening (16:00–18:00 h). The butterflies were observed feeding at the top of the inflorescence and visiting both buds ready to open and open flowers, and they could be considered marginal pollinators. Hummingbird moths were also observed visiting the flowers; they tended to visit the panicle apex first. The activity of honeybees, butterflies, and bumblebees was noticed concomitantly at the peak period of the flower opening, which suggests the possible role of these insects in pollination.

### 2.6. Reproductive Success in Different Crown Layers

The production of inflorescence and the production of flowers did not differ significantly between crown layers (Table 2). Fruit set within inflorescence showed a highly significant difference between crown layers (Table 2, *F* = 12.24, *p* = 0.0076) in trees. We have also noticed a trend of increasing the number of inflorescences and flower production from the lower canopy (LC) to the upper canopy layer (UP), which consequently enhanced the fruit set in the middle and upper canopy layers (Table 4). Surprisingly, there was a significant difference in reproductive success (proportion of the number of flowers converted into fruit) between crown layers in trees (Table 2, *F* = 14.238, *p* = 0.0053). Although there are six ovules in each ovary, successful pollination invariably leads to the formation of only one seed in a fruit. Only the ovule, located at either the first or second position towards the stylar end, matures into a seed. The remaining ovules aborted, resulting in very low reproductive potential in trees (0.167).

## 3. Discussion

This study reports the first comprehensive study of the floral biology and reproductive effort of *A. indica*. Many of the researchers have investigated the various reproductive aspects of the genus *Aesculus,* including the species *A. indica,* although many parts are still unknown. The andromonoecious sex expression in *A. indica* depicts a significant difference in the number of male flowers and perfect flowers in an inflorescence. The surplus production of male flowers maximizes pollinator attraction and functions as a pollen donor to fertilize another hermaphrodite flower, which enhances female and male fitness within the population. The study conducted by past workers stated that staminate flowers may have two selective advantages. First, they provide a pollen surplus that enhances male fitness [25,26,27,28]. Secondly, staminate flowers increase floral play and improve pollinator attendance, which in turn benefits female fitness [25,26,27,28,29]. Staminate flowers and perfect flowers were morphologically alike except for the incomplete development of pistils in staminate flowers. A more advanced study is required to understand the floral sexual dimorphism in this species.

In *A. indica*, a diurnal pattern of anthesis has been witnessed; the peak period of anthesis coincides with the maximum activity of the floral visitors. The flowering time of each species is genetically fixed, while it is highly variable with environmental factors, mainly precipitation, temperature, and relative humidity [30,31,32,33]. In addition, the flowering pattern of *A. indica* was found to be asynchronous, i.e., blooming of floral buds at different stages of development even on the same tree crown. Similar results were also reported for *Dalbergia sissoo* Roxb. [34], *Bombax ceiba* L. [35], and *Tectona grandis* L.f. [36]. Individuals with asynchronous flowering decrease reproductive output, the amount of pollen, the number of pollen donors, and the levels of outcrossing compared to individuals blooming during the same period [37].

There was no significant variation in pollen production between perfect and staminate flowers. Hence, it can be inferred that there would be a differential cost of expenditure for the production of staminate and perfect flowers; probably, the production of flowers serving only pollen sources would be less expensive. Therefore, the trees often produce more staminate flowers to increase pollen output and female fitness. Similar findings were also reported by Reale et al. [38]. The hermaphrodite (perfect) flowers possess protogynous dichogamy (the female phase preceding the male phase). Stigmas were receptive before the anthesis, and anther dehisced 5–6 h after the anthesis, indicating that the hermaphrodite flowers were well adapted to outbreeding and declined the chances of auto-pollination in *A. indica*. Protogynous dichogamy is one of the mechanisms promoting outcrossing in *Acacia* [39]. At anthesis, anthers were not dehisced, resulting in greater chances of viable pollen from outside depositing over the stigma and getting fertilized. Features like marked protogyny and inter-level asynchrony promote xenogamy; however, intra-level asynchrony may result in geitonogamy.

A high percentage of pollen viability from both hermaphrodite and staminate flowers exerts direct control over pollen-stigma interactions [40,41], fruit set, and gene flow [42]. Perhaps pollen germination was significant with time and concentration of germination media. The pollen germination increased with an increase in time until 5 h; later, there was a steady decline in the germination rate. 20% sucrose concentration exhibited maximum pollen germination, as sucrose is the carbohydrate source for pollen germination and tube growth [43]. The results on controlled pollination reveal predominantly self-incompatibility in *A. indica,* as no fruit set was observed from autogamous (self-pollination) pollination. Hence, pollination is completely dependent on vectors for its reproductive assurance. Interestingly, fruit set under open pollination was almost 10% less than that of open-cross pollination, which would be due to the absence of out-crossed pollen. Kalinganire et al. [44] mentioned that the poor fruit set following natural open pollination is mainly due to the absence of outcrossed pollen. This is probably due to low visitation rates by effective pollinators.

*A. indica* was found to possess all the characteristics of anemophily syndrome, such as long-exerted terminal inflorescence, low nectar, fewer flowers, a high P/O ratio, and small and smooth pollen produced in large numbers [45,46,47]. The trees produced pollen grains between one billion and eight billion per tree, which is generally the range reported for several temperate anemophilous broadleaved tree species [48]. Moreover, a significant amount of the pollen of *Aesculus hippocastanum* has been reported to occur in the air [4,49]. The existence of protogyny, asynchronous flowering, and self-incompatibility rules out the possibilities of self- (intrafloral or geitonogamous) pollination in the canopies. The present studies on floral visitors have revealed that the flowers of *A. indica* were frequently visited by five insect species from three orders, such as Hymenoptera, Thysanoptera, and Lepidoptera. It can be inferred that the pollination system in *A. indica* may have adapted to support both entomophily and anemophily, the occurrence of which is referred to as ambophily [50]. A similar finding was also observed for *Aesculus hippocastanum* [49]. However, verification of anemophily needs further experiments on pollen concentration in the air [51].

Most of the foragers were visiting the flowers during the peak period of blooming (08:00–10:00 h) of the day. It was apparent that anther dehiscence occurred 5–6 h later than the anthesis and did not coincide with the pollinator activity, clearly indicating that the foragers visit to the flower was inadvertent. The higher number of male flowers may reduce the probability of bisexual flower visits, consequently reducing the fruit set. Alternatively, pollen grains of 1-day-old opened flowers were contributing as pollen donors to freshly opened flowers, augmenting maximum cross-fertilization. Despite the frequent incidents of bees, thrips, and butterflies with flowers, there may also be increased chances of pollen theft, which negatively affects seed production [16,52,53,54]. Due to these reasons, insects cannot be considered effective pollinators in *A. indica*. Furthermore, frequent visits by more insects to the flowers increase the pollen concentration in the air, which assists wind pollination in *A. indica*. The assistance of insects in anemophily was recorded in *Phyllostachys nidulari* Munro [49], *Plantago lanceolata* L. [55], and *Mallotus* [50]. Therefore, the pollination syndrome of *A. indica* supports both wind (anemophily) and insect (entomophily) pollination.

We found that there was a significant difference in the proportion of flowers set into fruit (reproductive success) in the canopy layers. Reproductive success was found to be markedly higher in the upper canopy layers as compared to the lower layers. Exactly what causes the variability in fruit production in crown layers? Several factors affect fruit production: (1) the differential cost of expenditure on reproduction in crown layers results in inadequate translocation of resources; (2) ample amounts of light availability and temperature increase the photosynthetic rate and reproductive success; (3) pollen limitation arises from the preferential activity of pollinator; and (4) natural selection pressure. Similar findings were reported by a number of workers (e.g., fruit development depends strongly on photosynthesis and light availability [22,23,24]; competition by the surrounding neighborhood of plants for those resources [56,57,58]; zones of the canopy that receive more sunlight are expected to distribute more photosynthates locally, from the leaf to the adjacent fruit: local translocation of resources hypothesis [23,24,59].

In *A. indica*, the fruit set and seed set rates are the same due to the production of 1-seed fruits (dehiscent fruit). Fruit and seed production is very low when compared to the number of flowers and ovules. Similar findings were also recorded for *Gmelina arborea* [60,61]. Lee [62] and Guitia’n et al. [63] proposed several factors that may limit fruit production, including extrinsic causes (pollen limitation, herbivory, frost, etc.) and intrinsic causes (genotypes, stored resource content, etc.). The most common factor reported in the literature is resource limitation [64,65,66]. Furthermore, we speculated that the low rate of reproductive potential (S/O) was due to the existence of a high degree of ovule abortion. High rates of ovule abortion in multi-ovulated species are still conjectural. It could be argued that self-incompatibility and intra-ovule competition within the ovary would likely be the primary reasons for ovule abortion under pollen surplus conditions in *A. indica*. The plants also particularly abort ovules fertilized with pollen of low genetic quality to improve the mean seed quality [67].

## 4. Materials and Methods

### 4.1. Study Area

This study was conducted in the natural moist temperate forests of the Ranichauri area situated adjacent to the College of Forestry, Ranichauri, Tehri Garhwal (latitude 30°18′ N, longitude 78°24′ E, altitude between 1750–2100 m a.s.l.) in the state of Uttarakhand, North west Himalaya, India (Figure 2). The nearby vegetation consisted of temperate broad-leaved and coniferous species, viz. *Myrica esculenta* Buch.-Ham. ex D. Don, *Pyrus pashia* L., *Quercus leucotrichophora* A. Camus, *Rhododendron arboreum* Sm., *Cedrus deodara* (Roxb.) G. Don, and *Pinus roxburghii* Sarg. The study was carried out from April 2016 to November 2016. The study area typically experiences a moist temperate climate with chilled winters.

The mean monthly maximum and minimum temperature during the study period oscillated between 7.0 °C and 29.9 °C. The trees of *A. indica* were selected under varying microclimatic conditions. The soil texture varies from salty clay loam to sandy loam in the region where the study area was situated. The Typic Udorthent mountain soil of the region was moderately shallow, slightly stony, and excessively drained. The soils of this area are reported to be acidic to slightly alkaline, and the nutrient index of the soil was medium [68]. The trees for the study were chosen based on accessibility and convenience, and they were free from biotic pressure.

### 4.2. Floral Morphology and Anthesis

Ten inflorescences were randomly tagged from five different twigs of each sample tree. The inflorescences in the trees were selected based on its accessibility for frequent observations. The basic stages of flower development were defined by a detailed examination of the flowers. To examine anthesis, flowers were observed at 2 h intervals within a day between 06:00 and 16:00 h. During the whole period of study, the number of opened flowers was observed. Opened flowers were marked with a permanent marker to avoid errors due to duplication and over-counting. To elucidate the anther structure and morphological changes of the stigma, flowers were collected and observed under a microscope (Q-Tech 1400599 (Noida, India)). Regular phenological observations were made for individual sample trees.

### 4.3. Pollen Production

Pollen estimation was performed for both male flowers and perfect flowers. The pollen production per flower in trees was assessed using the noon loop method [36]. Anthers were obtained from closed flowers just prior to anthesis and were put into a test tube containing five drops of 5% glycerol, crushed with a glass rod, and pollen grains suspended in the test tube. The number of pollen grains was counted under a compound microscope (Olympus 703134). Pollen production per anther was determined with the formula P = (Tp/N) × n, where Tp is the summation of the total number of counted pollen grains in all the five drops, N is the total number of samples (slides) used for counting, and n is the total number of drops. The production of pollen grains per flower was estimated by multiplying the number of pollen grains per anther with the number of anthers per flower. The mean pollen production of a tree was determined by using the formula TP = N × F × A × P, where TP is the total pollen grains per tree, N is the number of inflorescences per tree, F is the average number of flowers per inflorescence, A is the average number of anthers per flower, and P is the average number of pollen grains per anther.

### 4.4. Stigma Receptivity and Pollen Ovule Ratio

Stigma receptivity was determined by immersing the stigmas of flowers in a 3% water solution of hydrogen peroxide; bubbling indicated peroxide activity and stigma receptivity [69]. Flowers from different inflorescences opened at almost the same time and were immersed at two-hour intervals up to thirty-eight hours to determine the average duration of stigma receptivity. Pollen-ovule (P/O) ratios were determined by dividing the number of pollen grains per flower by the number of ovules per flower [70]. To determine the ovule number, a cross-section of the ovary was made, and the number of ovules was counted. Twenty flowers were used to obtain the ovule number per flower. The ratio (P/O) was derived from the mean pollen quantity and the mean ovule quantity.

### 4.5. Pollen Viability

The viability of the pollen grains was tested in vitro by using the acetocarmine staining method [71]. We used anthers from fresh flowers before the anthesis. Pollen grains were dusted onto a clean microscopic slide, to which an equal amount of acetocarmine (0.5%) and glycerol was added and warmed gently. The slides were incubated for 10–15 min. Stained pollen grains within the microscopic slide were counted as viable pollen while shriveled; empty and weakly stained grains were recorded as non-viable.

### 4.6. Pollen Germination

Pollen germination of *A. indica* was tested separately for freshly collected pollen grains from both male and perfect flowers under in vitro conditions in sucrose solutions (germination media) with concentrations of 10%, 20%, and 30%. Pollen grains were placed in Petri dishes containing germination media (20 mL) and maintained at 20.4 °C (room temperature) for 48 h. Germinated pollen grains were observed under the binocular light microscope in each solution. The pollen grains were considered germinated when the pollen tube length was greater than the diameter of the pollen grain [72]. The germination rate was quantified as the percentage of germinated pollen grains per 100 evaluated grains.

### 4.7. Breeding System and Floral Visitors

A total of 100 flowers from each sample tree were subjected to four types of treatments (25 flowers for each treatment): (1) open pollination—the flowers were left without any intervention (no bag, no artificial pollination) and observed up to fruit set; (2) obligate self-pollination—flower buds were bagged throughout the flowering period and observed up to fruit set; (3) open cross-pollination—removing the anthers from the flowers before anthesis and was observed for fruit set; and (4) apomixes—removing both the anther and the stigma of the flowers in bud stage and observed for fruit set.

Pollinator observations were carried out throughout the day between 06:00 and 16:00 h on each experimental tree continuously for 8 days in May. Insects visiting the inflorescences were observed for their foraging behavior, the number of individuals, and the duration of the visit. The insects were captured using an insect trapping net (Sweep net) and polythene bags every hour during the foraging period of flower visitors (06:00 h to 18:00 h). Unidentified insects’ specimens were dried and placed into separate rectangular papers used for further identification. The double mounting method has opted for small insects.

### 4.8. Reproductive Success in Different Crown Layers

The crown length of the selected sample trees was measured using the Haga altimeter. The total crown length was divided into three equal proportions: the upper canopy layer (UC), middle canopy layer (MC), and lower canopy layer (LC) (Figure 3). The total number of inflorescences in each canopy layer was counted manually and multiplied by the average number of flowers per inflorescence to get the total number of flowers. Twenty inflorescences each in the upper, middle, and lower canopy were tagged in each sample tree and observed for fruit formation per inflorescence. The fruit set and seed set were calculated separately for each canopy layer. Here, fruit set is defined as the proportion of flowers that developed into fruits in each inflorescence. Seed set is defined as the proportion of ovules that develop into seeds in all the mature fruits within an inflorescence. Twenty inflorescences each in the upper, middle, and lower canopy were tagged in each sample tree and observed for fruit formation. The average fruit set in each canopy layer was calculated by multiplying the number of fruit sets per inflorescence by the total number of inflorescences in each canopy layer and expressed as a percentage. Similarly, seed sets per tree were calculated as the average number of seed sets per inflorescence multiplied by the total number of inflorescences within the canopy. The female reproductive success of a tree was calculated by multiplying the mean number of fruits per plant by the mean number of seeds per fruit [73].

### 4.9. Data Analysis

The mean ± standard deviations of the mean were calculated for all the measurements. The effect of time and tree-to-tree variation on flower opening was examined by two-factor analysis of variance (ANOVA), by using the Web Agri Stat Package, software, 2004 developed by the Indian Council of Agricultural Research (ICAR) Central Coastal Research Institute, Goa, India with time and tree-to-tree difference as fixed effects. Student’s *t*-test (test for mean difference) was performed to assess the difference in pollen production in staminate and perfect flowers and the number of staminate and perfect flowers within the inflorescence. The effect of time and sucrose concentration on pollen germination was also examined with a two-way ANOVA, with time and concentration of the solution as fixed effects.

Among crown layers, variation in the production of inflorescence, number of flowers, number of fruits per inflorescence, and reproductive success were assessed by a single-factor ANOVA with crown layers as a fixed effect. The effect of pollination treatments on fruit sets was also verified by a single-factor ANOVA with treatment as a fixed effect. The statistical analysis of the data was performed by Web Agri Stat Package, software, 2004 developed by the Indian Council of Agricultural Research (ICAR) Central Coastal Research Institute, Goa, India.

## 5. Conclusions

This study indicates that *A. indica* is a self-incompatible and outcrossing species. The surplus male flower production in *A. indica* is likely to preserve resources for increasing female fitness and function, as the male flowers substantially serve to attract pollinators and primarily act as pollen donors. Ambophily (anemophily as well as entomophily) is the primary pollination mechanism in *A. indica.* We found that crown layers have a significant impact on flower production, fruit, and seed set, presumably due to the differential cost of reproduction expenditure in crown layers. This study also conveys a novel insight into the reproductive response of *A. indica* under the current ecological state of the Garhwal Himalaya, which may affect the population dynamics of the species and be further helpful for the accomplishment of land management and species conservation programs associated with *A. indica*. The floral dimorphism, ovule abortion, and productivity of crown layers are the subject of subsequent study in *A. indica*.

## Figures and Tables

**Figure 1 plants-13-00183-f001:**
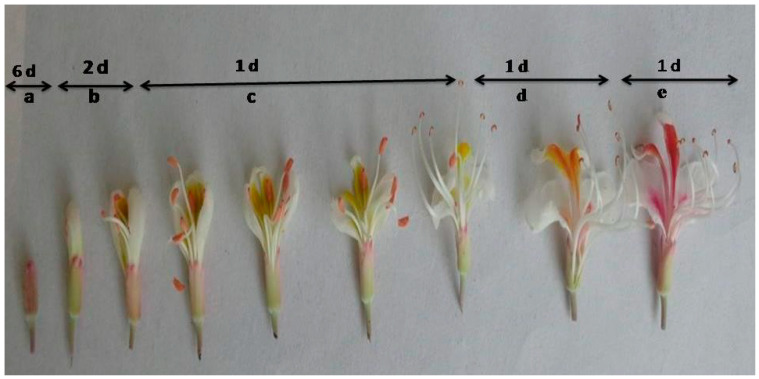
Individual flower developmental stages of *A. indica* and the mean number of days to reach each flower stage. The main stages represented in sequence are from left to right: (a) early bud to petal enlargement; (b) anthesis; (c, d) pre-fertilization stages; and (e) post-fertilization stages.

**Figure 2 plants-13-00183-f002:**
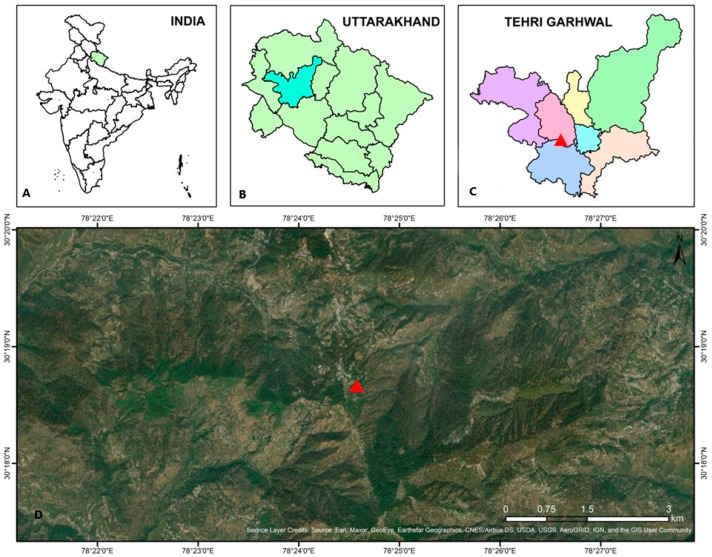
Map of study area. (**A**) Location of the state in country, (**B**) Location of the district in state, (**C**) Location of the study site in the district, (**D**) Geographical view of study area.

**Figure 3 plants-13-00183-f003:**
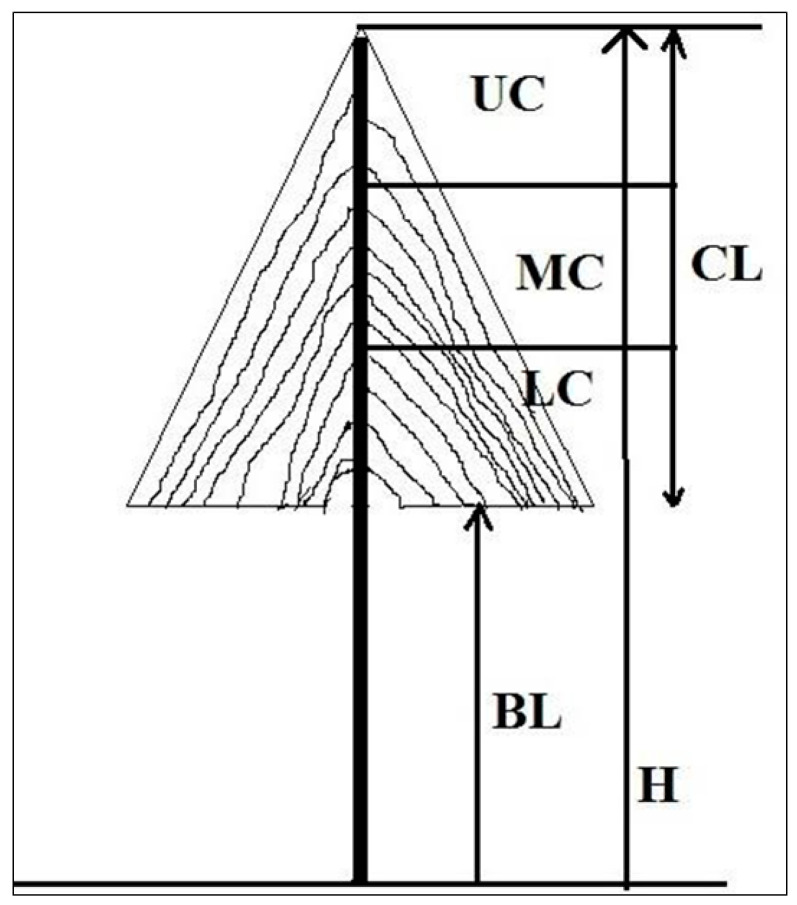
Diagrammatic sketch of crown differentiation for assessing reproductive success in crown layers. H: total height of the tree; BL: bole length; CL: crown length (H-BL); UC: upper crown layer; MC: middle crown layer; and LC: lower crown layer.

**Table 1 plants-13-00183-t001:** Anthesis in relation to the time of the day in *A. indica*.

Time (h)	TF (Mean ± SD)	% of Anthesis
06:00	1498.00 ± 45.36	51.58 ± 1.35
08:00	402.00 ± 22.37	13.82 ± 0.43
10:00	262.00 ± 37.95	8.99 ± 1.08
12:00	333.00 ± 6.37	11.46 ± 0.18
14:00	253.00 ± 5.88	8.70 ± 0.26
16:00	157.00 ± 15.62	5.42 ± 0.38

TF: total number of flowers opened; SD: standard deviation.

**Table 2 plants-13-00183-t002:** ANOVA of the effect of time and tree-to-tree difference on anthesis, the effect of time and concentration of the solution on pollen germination, the effect of crown layers on (production of inflorescence, number of flowers, fruit set per inflorescence, and reproductive success), and the effect of pollination treatment on fruit set.

Measured Variable	Independent Factor	df	*F*-Value	*p*-Value (*p* < 0.05)
Flower opening	1. Sample trees	2	3.260 ^NS^	0.81274
	2. Time	5	968.550 *	0.00001
Pollen germination (perfect flower)	Time	5	48.740 *	0.00001
	Concentration of solution	2	18.108 *	0.00047
Pollen germination (staminate flower)	Time	5	67.847 **	0.00001
	Concentration of solution	2	31.130 **	0.00005
Production of inflorescence	Crown layers	2	1.323 ^NS^	0.33405
Number of flowers	Crown layers	2	1.272 ^NS^	0.34618
Fruit set/inflorescence	Crown layers	2	12.239 *	0.00763
Reproductive success	Crown layers	2	14.238 *	0.00527
Fruit set	Pollination treatments	3	13.833 *	0.01566

** Significance at *p* < 0.01, * Significance at *p* < 0.05, ^NS^ = non-significant.

**Table 3 plants-13-00183-t003:** Insect visitors to *A. indica* recorded at the crop improvement research station, College of Forestry, Ranichauri, Uttarakhand, India, during 8-day observations in May.

Time (h)	Name of Visitors	Scientific Name	Order	Number of Individual	Duration of Visit
06:00–08.00	Bumble Bee	*Bombus* spp.	Hymenoptera	22	13–15 s
	Thrips		Thysanoptera	Abundant	
08:00–10:00	Bumble Bee	*Bombus* spp.	Hymenoptera	12	8–13 s
	Honeybee	*Apis mellifera*	Hymenoptera	7	4–8 s
	Thrips		Thysanoptera	Abundant	
	Himalayan Spotted Flat	*Celaenorrhinus munda*	Lepidoptera	5	7–13 s
10:00–12:00	Bengal Spotted Snow Flat	*Tagiades menaka*	Lepidoptera	4	6–16 s
	Thrips		Thysanoptera	Abundant	
	Honeybee	*Apis mellifera*	Hymenoptera	2	4–8 s
12:00–14:00	Bengal Spotted Snow Flat	*Tagiades menaka*	Lepidoptera	2	6–16 s
	Honeybee	*Apis mellifera*	Hymenoptera	4	4–8 s
	Thrips		Thysanoptera	Abundant	
14:00–16:00	Bengal Spotted Snow Flat	*Tagiades menaka.*	Lepidoptera	3	6–16 s
	Bumble Bee	*Bombus* spp.	Hymenoptera	9	8–13 s
	Thrips		Thysanoptera	Abundant	
	Honeybee	*Apis mellifera*	Hymenoptera	4	4–8 s
	Hummingbird Moth	*Hemaris* spp.	Lepidoptera	2	1–2 s
16:00–18:00	Bengal Spotted Snow Flat	*Tagiades menaka*	Lepidoptera	10	16 s

**Table 4 plants-13-00183-t004:** Reproductive success on different crown layers in sample trees of *A. indica*.

Observed Variables		*n* = 3	
	LC (m ± SD)	MC (m ± SD)	UC (m ± SD)
Total number of inflorescences	43.60 ± 25.60	81.00 ± 48.50	16.60 ± 69.50
Total flower production	17,672.00 ± 10,451.00	32,835.00 ± 19,791.00	51,270.00 ± 28,877.00
Total number of fruits/inflorescences	2.26 ± 1.90	2.90 ± 0.40	3.50 ± 0.30
Total number of fruits/trees	103.00 ± 63.00	255.00 ± 162.00	464.60 ± 261.00
% of fruit set	0.58 ± 0.04%	0.74 ± 0.07%	0.93 ± 0.81
Average weight of fruit (g)		29.30 ± 0.50	
Total weight of fruit/tree (g)	3012.00 ± 1834.00	7396.00 ± 4667.00	10,560.00 ± 5467.00
Average number of ovules/ovaries	6.00	6.00	6.00
Average number of seeds/fruits	1.00	1.00	1.00
Average number of seeds/trees		769.00 ± 461.00	
Reproductive success	0.0058	0.0074	0.0093
Reproductive potential	0.167	0.167	0.167

LC: lower canopy, MC: middle canopy, and UC: upper canopy (m ± SD: mean ± standard deviation).

## Data Availability

The data presented in this study are available on request from the corresponding author.

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
