# Peer review of "Effect of Crown Layers on Reproductive Effort and Success in Andromonoecious *Aesculus indica* (Wall. ex Camb.) Hook (Sapindaceae) in a Temperate Forest of Garhwal Himalaya"

_plants, 2024, doi:10.3390/plants13020183_

Round 1

Reviewer 1 Report

Comments and Suggestions for Authors

Dear authors, excellent work. It was a pleasure to review this manuscript. You will find minor corrections that need to be made in the paper in pdf format in the attachment. After minor revision, the paper can be published in the journal Plants.

Author Response

Reviewer 1

Comments

Dear authors, excellent work. It was a pleasure to review this manuscript. You will find minor corrections that need to be made in the paper in pdf format in the attachment. After minor revision, the paper can be published in the journal Plants.

Reply: The authors are highly grateful to the reviewer for providing a detailed and thorough review of the manuscript. All the suggestions and corrections have now been incorporated and the manuscript has been revised accordingly.

Reviewer 2 Report

Comments and Suggestions for Authors

The reviewed paper is devoted to characterization of selected reproductive features of Indian species of a walnut focused on differential position of inflorescences within a canopy. This topic is of interest for a broad range of specialist. However, the manuscript under discussion needs very serious elaboration before its acceptance for publication. Here are only major flaws outlined.

1.  First and foremost, both text and style of this text require a very serious elaboration. I strongly recommend that the authors should have their manuscript reviewed by either a native speaker or a specialized language service. In many cases, I could not manage to understand what the authors meant by their statements. I have corrected some points in first four pages but much more are left. Please see the manuscript file attached.
For example, the key term 'crown layers' in some places should be used in some other way. In line 449, the authors state that 'crown layers have a significant impact on flower production'. Actually, layers themselves do not impact on anything while the position in a canopy does.
The description of flower and inflorescence morphology is lacking conventional terms used in plant morphology and needs to be revised. Numerous corrections are needed in the statistical part.
2. The part devoted to visitations by insects seems unrelated to the main scope of this paper, i.e. differences in reproductive success between canopy layers. Besides this, the very fact of some insects visiting flowers does not indicate they pollinate them. Very many insects can feed on flowers acting as pollen/nectar thieves. It is a very challenging task to unambiguously prove that certain insect species pollinate certain plant efficiently. I would rather the authors either completely omitted this entomological part or rephrased it to make all conclusions more cautious.
3. The Discussion part is somewhat shaky and rests on debatable assumptions. Some of the points highlighted by the authors seem trivial. For example, there is nothing exceptional in the fact that flowers in a tree open within a certain period, not simultaneously. It is more a product of inflorescence morphology rather than of a life form. Herbaceous plants possessing cymose inflorescences (like many of Boraginaceae) would also exhibit a prolonged flowering like Aesculus. It is also unclear how natural selection pressure could affect fruit production within a single canopy.

To sum up, I recommend that this manuscript should be rejected with an encouragement to resubmit after a deep revision. However, the authors should be aware that the efficient review could be done only after both language and style of this manuscript are revised. I also wish the authors good luck.

Comments on the Quality of English Language

Both language and style of this paper need to be thoroughly revised before resubmission.

Author Response

Reviewer 2

Comments

The reviewed paper is devoted to characterization of selected reproductive features of Indian species of a walnut focused on differential position of inflorescences within a canopy. This topic is of interest for a broad range of specialist. However, the manuscript under discussion needs very serious elaboration before its acceptance for publication. Here are only major flaws outlined.

  1. First and foremost, both text and style of this text require a very serious elaboration. I strongly recommend that the authors should have their manuscript reviewed by either a native speaker or a specialized language service. In many cases, I could not manage to understand what the authors meant by their statements. I have corrected some points in first four pages but much more are left. Please see the manuscript file attached.
    For example, the key term 'crown layers' in some places should be used in some other way. In line 449, the authors state that 'crown layers have a significant impact on flower production'. Actually, layers themselves do not impact on anything while the position in a canopy does.
    The description of flower and inflorescence morphology is lacking conventional terms used in plant morphology and needs to be revised. Numerous corrections are needed in the statistical part.

Author’s remark: Thank you for the valuable suggestions and critical comments. The authors have corrected the language of the manuscript to the best of their knowledge. Most of the suggestions recommended by the reviewer are also incorporated in the manuscript now. In the statistical part, in response to the query of the reviewer regarding students’s t test, the authors would like to mention that the Student's t-test is generllay used when the sample size is possibly small i.e less than approximately 30.

                        Kindly refer to the following chapter for the same:

Andrew P. King, Robert J. Eckersley. Inferential Statistics II: Parametric Hypothesis Testing, Editor(s): Andrew P. King, Robert J. Eckersley, Statistics for Biomedical Engineers and Scientists, Academic Press, 2019, Pages 91-117, https://doi.org/10.1016/B978-0-08-102939-8.00014-1.

  1. The part devoted to visitations by insects seems unrelated to the main scope of this paper, i.e. differences in reproductive success between canopy layers. Besides this, the very fact of some insects visiting flowers does not indicate they pollinate them. Very many insects can feed on flowers acting as pollen/nectar thieves. It is a very challenging task to unambiguously prove that certain insect species pollinate certain plant efficiently. I would rather the authors either completely omitted this entomological part or rephrased it to make all conclusions more cautious.

Author’s remark: Thank you for the worthy suggestion. We have rephrased the conclusive statements in the entomological part of the study.

  1. The Discussion part is somewhat shaky and rests on debatable assumptions. Some of the points highlighted by the authors seem trivial. For example, there is nothing exceptional in the fact that flowers in a tree open within a certain period, not simultaneously. It is more a product of inflorescence morphology rather than of a life form. Herbaceous plants possessing cymose inflorescences (like many of Boraginaceae) would also exhibit a prolonged flowering like Aesculus. It is also unclear how natural selection pressure could affect fruit production withina single canopy.

Author’s remark: We are greatly thankful to the reviewer for valuable comments and suggestions. The part of discussion that is being considered as insignificant by the esteemed reviwer has been mentioned only to support some of the observations of the study. Moreover; these statements are also backed by various previous scientific studies in the same context.

  1. To sum up, I recommend that this manuscript should be rejected with an encouragement to resubmit after a deep revision. However, the authors should be aware that the efficient review could be done only after both language and style of this manuscript are revised. I also wish the authors good luck.

Author’s remark: As recommended by the reviewer we have made all the possible corrections in the manuscript for better presentation of the study.

Reviewer 3 Report

Comments and Suggestions for Authors

Dear authors,

this study is interesting with scientific and practical importance. The manuscript is about the effect of crown layers on floral biology and reproductive effort of Aesculus indica (Wall. ex Camb.) Hook. The authors proved that the crown layers have a significant impact on flower production, fruit, and seed set.

In the manuscript, introduction and objectives are well and clear written. The materials and methods are given in details. The results obtained and presented in 2 figures and 4 tables are relevant to the proposed objectives. The discussion is appropriate in the context of the results. The conclusions are supported by the results. The references are representative in the field of study.

Before accepting of the manuscript, following parts have to be corrected:

line

41-43   check whether Aesculus  indica falls under the section Calothyrsus or Aesculus

56        Dirr,    >          Dirr

58        t o   h a v e      >          to have

93        [22, 23, 24]     >          [22-24]

97        t h e     >          the

135      during >          During

140      Pollen estimation        >          and pollen estimation

158      [ 2 7 .   >          [27].

167      [ 2 9 ] .             >          [29].

265-266          TF:  Total  number  of flowers opened, n= number of focal trees, and sd: standard deviation.            >          move below the table

270      **Significance at P<0.1 *Significance at P<0.05, NS= Non-significant.   

>          move below the table

295      (83.33%  for perfect flowers and 84.19% for male flowers) 10% sucrose.

            >          83.33% for perfect flowers and 84.19% for male flowers in 10% sucrose.

298      P=0.00156      >          in Table 2, P= 0.01566

320      262crown        >          262 crown

320      Table   2,   F= 1.323,   P=0.33341,   F=1.272,   P=0.3462

            >          Table   2,   F= 1.323,   P=0.3341,   F=1.272,   P=0.3462

327      Table  2,  F=14.237, P=0.0052          >          Table  2,  F=14.238, P=0.0053

332-333          LC:  Lower canopy, MC:  Middle canopy, and UC: Upper canopy, (m ± sd: mean ± standard deviation).

            >          move below the table

344      [32, 33, 34, 35]           >          [32-35]

346      [ 3 2 , 3 3 ,  3 6 ]         >          [32,33,36]

352-353          [37, 38, 39,40]            [37-40]

353      A. indica         >          A. indica

355      [41       >          [41]

364      [ 4 4 ]  >          [44]

369      [ 4 5 ]  >          [45]

375      [46, 47]           >          [46,47]

383      10  %   >          10%

390      [51, 52, 53] .   >          [51-53].

392      [ 5 4 ] .            >          [54].

393      esculus            >          Esculus

400      [56]).   >          [56].

401      [ 5 5 ] .            >          [55].

412-413          [16, 58, 59, 60]           >          [16,58-60]

417      [ 6 1 ] ;            >          [61];

417      [ 6 2 ] ;            >          [62];

429      [22, 23, 24] ;   >          [22-24];

430      [63,  64]          >          [63,64]

432      [ 2 3 ,   6 6 ,   2 4 ] .    >          [23,24,66].

436      [ 6 7 ,   6 8 ] .  >          [67,68].

436      [ 6 9 ]              >          [69]

436      [ 7 0 ]              >          [70]

439-440          [e.g.  71, 72]    >          [71,72]

457      Conceptualization, P.P., A.S,            >          Conceptualization, P.P., A.S.

458 2x             M.K.R             >          M.K.R.

459      A.S;     >          A.S.;

460      M.K., and       >          M.K. and

461      supervision,  V.P.K;   >          supervision, V.P.K.;

470-602          the list of references has to be revised according to the Instructions                 for Authors

Comments on the Quality of English Language

Minor editing of English language required.

Author Response

Reviewer- 3

Comments

Dear authors,

This study is interesting with scientific and practical importance. The manuscript is about the effect of crown layers on floral biology and reproductive effort of Aesculus indica (Wall. ex Camb.) Hook. The authors proved that the crown layers have a significant impact on flower production, fruit, and seed set.

In the manuscript, introduction and objectives are well and clear written. The materials and methods are given in details. The results obtained and presented in 2 figures and 4 tables are relevant to the proposed objectives. The discussion is appropriate in the context of the results. The conclusions are supported by the results. The references are representative in the field of study.

Before accepting of the manuscript, following parts have to be corrected:

Line 41-43   check whether Aesculus  indica falls under the section Calothyrsus or Aesculus

56        Dirr,    >          Dirr

58        t o   h a v e      >          to have

93        [22, 23, 24]     >          [22-24]

97        t h e     >          the

135      during >          During

140      Pollen estimation        >          and pollen estimation

158      [ 2 7 .   >         [27].

167      [ 2 9 ] .             >          [29].

265-266          TF:  Total  number  of flowers opened, n= number of focal trees, and sd: standard deviation.            >          move below the table

270      **Significance at P<0.1 *Significance at P<0.05, NS= Non-significant.   

>          move below the table

295      (83.33%  for perfect flowers and 84.19% for male flowers) 10% sucrose.

            >          83.33% for perfect flowers and 84.19% for male flowers in 10% sucrose.

298      P=0.00156      >          in Table 2, P= 0.01566

320      262crown        >          262 crown

320      Table   2,   F= 1.323,   P=0.33341,   F=1.272,   P=0.3462

            >          Table   2,   F= 1.323,   P=0.3341,   F=1.272,   P=0.3462

327      Table  2,  F=14.237, P=0.0052          >          Table  2,  F=14.238, P=0.0053

332-333          LC:  Lower canopy, MC:  Middle canopy, and UC: Upper canopy, (m ± sd: mean ± standard deviation).

            >          move below the table

344      [32, 33, 34, 35]           >          [32-35]

346      [ 3 2 , 3 3 ,  3 6 ]         >          [32,33,36]

352-353          [37, 38, 39,40]            [37-40]

353      A. indica         >          A. indica

355      [41       >          [41]

364      [ 4 4 ]  >          [44]

369      [ 4 5 ]  >          [45]

375      [46, 47]           >          [46,47]

383      10  %   >          10%

390      [51, 52, 53] .   >          [51-53].

392      [ 5 4 ] .            >          [54].

393      esculus            >          Esculus

400      [56]).   >          [56].

401      [ 5 5 ] .            >          [55].

412-413          [16, 58, 59, 60]           >          [16,58-60]

417      [ 6 1 ] ;            >          [61];

417      [ 6 2 ] ;            >          [62];

429      [22, 23, 24] ;   >          [22-24];

430      [63,  64]          >          [63,64]

432      [ 2 3 ,   6 6 ,   2 4 ] .    >          [23,24,66].

436      [ 6 7 ,   6 8 ] .  >          [67,68].

436      [ 6 9 ]              >          [69]

436      [ 7 0 ]              >          [70]

439-440          [e.g.  71, 72]    >          [71,72]

457      Conceptualization, P.P., A.S,            >          Conceptualization, P.P., A.S.

458 2x             M.K.R             >          M.K.R.

459      A.S;     >          A.S.;

460      M.K., and       >          M.K. and

461      supervision,  V.P.K;   >          supervision, V.P.K.;

470-602          the list of references has to be revised according to the Instructions for Authors.

Author’s remark: The authors are highly grateful to the reviewer for providing a detailed and thorough review of the manuscript. All the suggestions and corrections have now been incorporated and the manuscript has been revised accordingly.

Round 2

Reviewer 2 Report

Comments and Suggestions for Authors

As I may see, the authors have made a certain revision of their manuscript following my and other reviewers' comments and suggestions. Some details of the methods applied are now clearer. However, this paper still needs an extensive language and style editing.
For example, the very first word of this article's abstract, 'andromonoecious', is an adjective, not a noun. That is why it cannot be a sex-expression and the whole sentence should be rewritten like, 'Andromonoecy is an unusual...'
In line 42, the generic name should be italicized. Moreover, as discussed in the first review, Hippocastanaceae as a family is not accepted anymore; the authors have corrected it in their title but this name still exists in the text. All generic and specific Latin names should be italicized. After the first mention, a generic name should be abbreviated till Ae., not A., however, placing this abbreviation to the very first position in a sentence should be avoided.
All these things were already discussed in the first review but the authors seem to have missed them. Although they do not influence the overall value of this paper, they do impact on the way how this text is perceived and appreciated. The authors should note that all minor flaws like these, especially of a biological kind, will not be corrected by editors and can persist in the final version of this paper.
In all Tables, all identical values, both average and s.d., should be given with equal precision. That is why it is not recommended to have F-values in the same column of Table 2 like 3.26, 18.108, and 13.8333; these all need to be given with an equal number of decimals. I guess, ** denotes p < 0.01, not 0.1, in Table 2's notes.
It is unclear why reproductive success and reproductive potential in Table 4 are given without units, with different number of decimals and withoud SD. Why are values of the reproductive potential missing in columns of LC and UC? Why are there empty cells in LC and UC columns in lines of average numbers? Are these data missing?
Unfortunately, the authors seem to have ignored my suggestion to improve their diagrammatic sketch in Fig. 2. Its quality is still worth improving.
Numerous flaws like these severely complicate the ability to get into the scientific content of the paper, not to mention that they impact on the authority of the journal negatively. I strongly recommend the authors to look through their paper thoroughly and do their best to improve it.

Comments on the Quality of English Language

The quality of English still deserves improving.

Author Response

  1. As I may see, the authors have made a certain revision of their manuscript following my and other reviewers' comments and suggestions. Some details of the methods applied are now clearer. However, this paper still needs extensive language and style editing.
    For example, the very first word of this article's abstract, 'andromonoecious', is an adjective, not a noun. That is why it cannot be a sex expression and the whole sentence should be rewritten like, 'Andromonoecy is an unusual...'

Author’s remark: Thank you for the worthy suggestion. The first statement of the abstract has now been rewritten following the suggestion of the reviewer. Also, the style and language editing in the manuscript has been done as suggested by the reviewer.

  1. In line 42, the generic name should be italicized.

Author’s remark: Required correction in line no. 42 has been made.

  1. Moreover, as discussed in the first review, Hippocastanaceae as a family is not accepted anymore; the authors have corrected it in their title but this name still exists in the text.

Author’s remark: Thank you for pointing out the required modification in the manuscript. The family name has been corrected throughout the manuscript.

  1. All generic and specific Latin names should be italicized.

Author’s remark: The suggested correction has been incorporated.

  1. After the first mention, a generic name should be abbreviated till , not A., however, placing this abbreviation to the very first position in a sentence should be avoided. All these things were already discussed in the first review but the authors seem to have missed them. Although they do not influence the overall value of this paper, they do impact the way this text is perceived and appreciated. The authors should note that all minor flaws like these, especially of a biological kind, will not be corrected by editors and can persist in the final version of this paper.

Author’s remark: Thank you for the kind suggestion but the authors would like to mention that an abbreviation of the generic name like Aesculus can be written as A.

Kindly refer to the following articles already published in mdpi:

Dridi, A.; Reis, F.S.; Pires, T.C.S.P.; Calhelha, R.C.; Pereira, C.; Zaghdoudi, K.; Ferreira, I.C.F.R.; Barros, L.; Barreira, J.C.M. Aesculus hippocastanum L.: A Simple Ornamental Plant or a Source of Compelling Molecules for Industry? Separations 202310, 160. https://doi.org/10.3390/separations10030160

Riaz, M.; Suleman, A.; Ahmad, P.; Khandaker, M.U.; Alqahtani, A.; Bradley, D.A.; Khan, M.Q. Biogenic Synthesis of AgNPs Using Aqueous Bark Extract of Aesculus indica for Antioxidant and Antimicrobial Applications. Crystals 202212, 252. https://doi.org/10.3390/cryst12020252

Zdravković-Korać, S.; Milojević, J.; Belić, M.; Ćalić, D. Tissue Culture Response of Ornamental and Medicinal Aesculus Species—A Review. Plants 202211, 277. https://doi.org/10.3390/plants11030277

  1. In all Tables, all identical values, both average and s.d., should be given with equal precision. That is why it is not recommended to have F-values in the same column of Table 2 like 3.26, 18.108, and 13.8333; these all need to be given with an equal number of decimals. I guess, ** denotes p< 0.01, not 0.1, in Table 2's notes.

Author’s remark: Thank you for the kind suggestion. The required corrections in the tables have been made.

  1. It is unclear why reproductive success and reproductive potential in Table 4 are given without units, with different numbers of decimals and without SD. Why are values of the reproductive potential missing in columns of LC and UC? Why are there empty cells in LC and UC columns in lines of average numbers? Are these data missing?

Author’s remark: The reproductive success of the tree was calculated as the proportion of the number of flowers converted into fruit and the reproductive potential is also the ratio of the number of ovules per ovary and the number of ovules that converted into seed. That is why there is no unit for the values of reproductive success and reproductive potential depicted in the table.

The missing values of reproductive potential have now been added to the table. In the lines of average number (Average number of seed/ tree, Average weight of fruit (g)), the single value written in the row is for all three layers as it is the average per tree.

  1. Unfortunately, the authors seem to have ignored my suggestion to improve their diagrammatic sketch in Fig. 2. Its quality is still worth improving.
    Numerous flaws like these severely complicate the ability to get into the scientific content of the paper, not to mention that they impact the authority of the journal negatively. I strongly recommend the authors look through their paper thoroughly and do their best to improve it.

Author’s remark: The authors like to extend sincere gratitude to the reviewer for his/her kind suggestions and critical comments for improving the quality of the manuscript. We have revised the manuscript thoroughly following all the suggestions of the esteemed reviewer.